# Mechanical Performance, Microstructure, and Porosity Evolution of Fly Ash Geopolymer after Ten Years of Curing Age

**DOI:** 10.3390/ma16031096

**Published:** 2023-01-27

**Authors:** Ikmal Hakem A. Aziz, Mohd Mustafa Al Bakri Abdullah, Rafiza Abd Razak, Zarina Yahya, Mohd Arif Anuar Mohd Salleh, Jitrin Chaiprapa, Catleya Rojviriya, Petrica Vizureanu, Andrei Victor Sandu, Muhammad FaheemMohd Tahir, Alida Abdullah, Liyana Jamaludin

**Affiliations:** 1Centre of Excellence Geopolymer & Green Technology (CEGeoGTech), Universiti Malaysia Perlis (UniMAP), Perlis 01000, Malaysia; 2Faculty of Chemical Engineering Technology, Universiti Malaysia Perlis (UniMAP), Perlis 01000, Malaysia; 3Faculty of Civil Engineering Technology, Universiti Malaysia Perlis (UniMAP), Perlis 01000, Malaysia; 4Synchrotron Light Research Institute, Muang, Nakhon Ratchasima 30000, Thailand; 5Faculty of Material Science and Engineering, Gheorghe Asachi Technical University of Iasi, 41 D. Mangeron St., 700050 Iasi, Romania; 6Faculty of Mechanical Engineering Technology, Universiti Malaysia Perlis (UniMAP), Perlis 01000, Malaysia

**Keywords:** fly ash geopolymer, compressive strength, porosity, crystallization

## Abstract

This paper elucidates the mechanical performance, microstructure, and porosity evolution of fly ash geopolymer after 10 years of curing age. Given their wide range of applications, understanding the microstructure of geopolymers is critical for their long-term use. The outcome of fly ash geopolymer on mechanical performance and microstructural characteristics was compared between 28 days of curing (FA28D) and after 10 years of curing age (FA10Y) at similar mixing designs. The results of this work reveal that the FA10Y has a beneficial effect on strength development and denser microstructure compared to FA28D. The total porosity of FA10Y was also lower than FA28D due to the anorthite formation resulting in the compacted matrix. After 10 years of curing age, the 3D pore distribution showed a considerable decrease in the range of 5–30 µm with the formation of isolated and intergranular holes.

## 1. Introduction

Geopolymer material has engineering qualities that are similar to OPC concrete [1], depending on the type of precursor used. Moreover, geopolymer is a supplementary cementitious material that has received significant attention in recent decades [2,3]. The compositions of the silica/alumina sources and the type of alkaline activators have a substantial influence on the properties of geopolymer. These properties can be initiated by using locally available precursors and the optimal activator dosage in geopolymer synthesis. Several countries are conducting research on geopolymer materials. The tremendous application possibilities have sparked increased interest in this technology, which is considered relatively innovative by current binder standards. However, its industrial spread and acceptability require more research into the various variables involved in its processing and service life.

According to De Vargas et al. [4], it is important to evaluate the various factors that are critical to ensuring effective polymerization of the raw material, such as reactive (amorphous) silicon concentration, glassy state level, and particle size distribution. The performance of the geopolymer structure will be mostly determined by the availability of ions in the solution to form a chemical bond. Ions from silicon, aluminum, and free oxygen coordinately bond during the reaction to produce inorganic polymer chains [5]. Sekou et al. [6] said that mixing suitable amounts of amorphous silica/alumina with the alkaline activator stimulates particle disintegration due to the high alkalinity in the medium, resulting in lixiviation and subsequent reactivity of the Al and Si ions. The coordinated Si–Al bonds formed with O are then exploited to generate monomers, which condense and precipitate as a gel. The final stage occurs when the majority of the available material in the reaction has organised into polymer(silates). When large proportions of inorganic polymers are generated and have hardened the geopolymer structure, the reaction is interrupted.

In term of mechanical performance, several experiments have been conducted in order to establish the durability performance of geopolymer material. The durability of the structure in an open environment is affected by the microstructure and crystalline evolution. According to Wong et al. [7], curing times up to 91 days have been demonstrated to boost strength even further. The dominance of calcium aluminium silicate hydrate (C–A–S–H) gel in the geopolymer system is related to the development strength. The curing temperature has an impact on the cement’s quality and durability [8]. Santa et al. [5] studied the microstructural evaluation in geopolymer samples after 8 years of curing age. Over the years, no structural degradation and denser microstructure were observed for the 8 years samples, hence, it may be related to long-term durability.

There are certain limitations to the traditional scanning electron microscopy (SEM) approach, which only offers pore shape assumptions and phase connectivity information in 2D images [9]. Otherwise, if standard X-ray tomography (µ-CT) is utilized, it is difficult to obtain a highlight-connected pore network for geopolymer material from the segmented pore [10]. Synchrotron X-ray tomography (SXTM) produces a high-resolution three-dimensional picture of pores and network arrangement [11]. Evaluating the microstructural properties of geopolymer in the short and long term is critical for identifying potential changes of structure degradation. More knowledge on these materials can help ensure their use in building projects as well as other fields that require new binders with distinct qualities.

An enormous number of works on geopolymer materials in the literature strive to understand the early phase of their development. In most situations, the samples were examined after ninety days of curing [12,13,14,15]. A deeper knowledge of the various alterations that occur in geopolymers following synthesis is critical in this context. Hence, certain samples were produced and preserved for 10 years to increase knowledge of geopolymer structure. Consequently, the aim of this work is to perform a mechanical and microstructural examination of a geopolymer sample in the early phases of curing (28 days), as well as samples of geopolymers preserved for more than 10 years.

## 2. Materials and Methods

The fly ash for this work was supplied by the Manjung Power Station in Lumut, Perak, Malaysia. X-ray fluorescence (XRF) was used to determine the chemical composition of the fly ash, as tabulated in Table 1. Due to the calcium oxide (CaO) level being greater than 10%, it can be categorized as Class-C fly ash according to ASTM C618-08 [16]. The fly ash was activated with an alkaline activator made by combining a technical grade sodium silicate (N_a2_SiO_3_) solution with sodium hydroxide (NaOH). The Na_2_SiO_3_ chemical composition was SiO_2_ = 30.1%, Na_2_O = 9.4%, and H_2_O = 60.5%, and the modulus ratio was 2 (where MS = SiO_2_/Na_2_O). The NaOH of 12 M was made by combining 97–99% pure sodium hydroxide pellets with distilled water. The alkaline activator was created by combining a 1:1 constant mass ratio of Na_2_SiO_3_ with 12 M NaOH solution.

### 2.1. Mixing of the Geopolymer Aggregate

In a 5 L quantity, the fly ash was mixed with the advanced prepared alkaline activator liquid for around 5 min to fabricate the geopolymer aggregate. The slurry was then poured into 100 mm × 100 mm × 100 mm cubic plastic molds, which were the dimensions of the samples. The specimens were then vibrated for 2 min on a vibrating table to expel any remaining air bubbles. During the hardening process, the geopolymer samples were coated with a thin coating of polyethylene to prevent water evaporation and then maintained in the laboratory for 24 h at ambient temperature before demolding. After demolding, the samples were coated with a thin layer of polyethylene for 28 days prior to measuring the compressive strength. Samples with the similar mixing design were prepared and set aside for curing at different time intervals. For each mixing design, the different curing period of fly ash-based geopolymer was investigated. Hence, two designs were synthesized. For instance, FA28D refers to the samples cured at 28 days, while FA10Y refers to the samples cured for 10 years. All the samples were stored in the laboratory at room temperature until the testing days.

### 2.2. Characterization Techniques

#### 2.2.1. Compressive Measurement and Microstructural Characterization

To determine the ultimate strength of the geopolymer, a compressive strength test was performed on a geopolymer aggregate sample in line with BS 1881-116:1983. The sample loaded had a force of 50.00 kN (11.24 Kips) and a loading speed of 5.00 mm/min (0.20 in./min). The loading pace rate was 0.1 kN/s (22.48 lb/s).

Using a JSM-6460LA model Scanning Electron Microscope (JOEL, Peabody, MA, USA) equipped with a secondary electron detector and energy-dispersive X-ray spectroscopy, the microstructural changes of the fly ash geopolymer were investigated. The fracture sample was cut into small pieces prior to the analysis. Palladium was used to coat the samples. The microstructure investigation was performed at 10 kV acceleration voltages and a working distance of 10 mm.

#### 2.2.2. X-ray Diffraction (XRD) Analysis

Shimadzu X-ray diffractometer XRD-6000 was used for phase analysis. The specimen was used for phase analysis. The specimen for analysis was processed as a powder. The XRD examination was carried out with Cu K radiation (λ = 1.5418 A) at 40 kV, 30 mA, at 2θ ranging between 10° and 80° with a scan rate of 1°/min. X’pert High score Plus software was used to analyse the XRD pattern (Columbia City, IN, USA).

#### 2.2.3. Synchrotron Micro X-ray Fluorescence (µ-XRF) and X-ray Tomographic Microscopy (SXTM)

The elemental distribution of the fly ash geopolymer was determined by Synchrotron-based µ-XRF. The experiments were carried out at Beamline 7.2 W, Synchrotron Light Research Institute (SLRI), Thailand. At BL 7.2 W, the synchrotron radiation was produced from a 6.5 T superconducting wavelength shifter. With Si(111) double crystal monochromator, 12 keV of the monochromatic X-ray beam was exploited in the experiments. The polycapillary half-lens, used to focus the X-ray beam, delivered a micro X-ray beam with the size of 25 × 25 μm^2^ onto the specimen in which the incident micro X-ray beam was perpendicular to the specimen (Bangkok, Thailand). 

The single element Vortex EM-650 silicon drift detector was placed was placed 45 degrees to the specimen to collect fluorescent X-rays emitted from the specimen, while the ccd camera, positioned parallel to the beam and perpendicular to the specimen, was used for area selection of the specimen. For this experiment, 0.3 × 0.3 mm^2^ of the region of interest on the sample was selected, and the scanning step 30 μm in horizontal and 30 μm in vertical was selected for raster scanning. Thus, the elemental distribution maps were collected using a 68 μm (h) × 35 μm (v) pixel size. The dwell time for each point was 30 s. The XRF spectra were analysed using PyMCA version 5.6.5 [17], which was developed by the Software Group of the European Synchrotron Radiation Facility (ESRF) (http://pymca.sourceforge.net/PyMca, accessed on 31 August 2020).

To prevent displacement and dehydration during tomographic scanning, each sample was placed in a cylindrical sample holder filled with cotton soaked in formaldehyde solution. SXTM experiments were carried out at the Synchrotron XTM beamline (BL 1.2 W: X-ray imaging and tomographic microscopy), Synchrotron Light Research Institute, Thailand. SXTM imaging of the fly ash geopolymer was performed at a distance of 34 m from the source using a filtered polychromatic X-ray beam with a maximum area of 4 × 8 mm^2^ at 10 keV [18]. The sample projections had a 200-µm-thick YAG:Ce scintillator, a white beam microscope (Optique Peter, France), and a pco.edge 5.5 sCMOS camera (2560 × 2160 pixels, 16 bits). The entire tomographic scan was taken at a pixel size of 1.44 µm, which offered a field of views (3.1 × 3.7 mm^2^). To identify the fine features of the fly ash geopolymer, two reconstruction tomographic volume scans were performed. The first scan was acquired over 180°, and the second scan was captured on the vertical axis of rotation as displaced parallel and horizontal to the actual camera [19]. The X-ray projection was then normalized by flat-field correction, patched, and reconstructed with Octopus software [20]. Drishti software was used to create 3D images of tomographic volume.

## 3. Results and Discussion

### 3.1. Compressive Strength Measurement

Figure 1a depicts the compressive strength of fly ash geopolymer aggregates at various fly ash/aggregate ratios at 28 days of curing. The results exemplified that the fly ash/aggregate ratios of 30/70 obtained the highest compressive strength (49.3 MPa) after 28 days of curing. When the fly ash content is too low (20% of dry weight) or too high (50–40% of dry weight), the compressive strength is insufficient. This suggests that high aggregate content in the geopolymer mixture does not guarantee high strength production. The presence of total fly ash further promotes the geopolymerization process. However, fly ash content less that than 20% causes the geopolymerization reaction to be less efficient, resulting in lower strength. Incorporation of aggregate and calcium source materials developed the compressive strength.

In order to evaluate the effect of curing age on the mechanical performance of the 28 days (FA28D) and 10 years (FA10Y), the highest fly ash/aggregate ratio (30/70) was selected for further investigations. After 10 years of curing age, the strength of the materials increased slightly as shown in Figure 1b. The FA10Y samples recorded the higher compressive strength (87.32 MPa), and the FA28D displayed the lower compressive strength (49.3 MPa). The impressive enormous rise in strength was mostly due to an increase in curing time for highly hardened samples, which played a vital role in the higher strength. The formation of a compacted microstructure will be discussed in the next section.

### 3.2. SEM

To determine the microstructure of fly ash geopolymer samples after 28 days and 10 years curing age, the fractured pieces were subjected to SEM/EDS analysis (Figure 2a–f). For samples at 28 days curing age (Figure 2a–c), fly ash particles were clearly unreacted and did not participate in the polymerization. Additionally, cracks and voids were also obtained indicating a permeable of geopolymer samples as depicted in Figure 2b. Cracks and voids were also discovered, which suggested that the geopolymer samples were porous. Furthermore, a micro crack in the microstructure pointed to inappropriate bonding of the geopolymer aggregate.

The morphologies of hardened samples after 10 years curing age are shown in Figure 2d–f. The samples have smaller pores than 28-day samples as seen in Figure 2e. The quality and durability of the samples are influenced by the curing temperature. In order to maintain uniform hydration, curing must be performed at temperatures that allow the moisture content to be sustained close to saturation. According to Lee et al. [21], long-term durability could be related to the properties and quantities of the pores present in the materials. The geopolymer binder contains pores that allowed liquid or gaseous agent to enter, leading to corrosion and shortening the material’s useful life. Hence, it is crucial to emphasize that a material’s curing age impact decreases as it become more durable.

The SEM images with larger magnification (5000×) depict an obvious crystallite formation in 10 years curing age samples with less crack and voids, respectively. In the amorphous geopolymer matrix, crystallites formed after prolonged curing age. Upon curing age up to 10 years, columnar anorthite crystal and plerospheres of thin-walled spherical fly ash [22] can be clearly seen in Figure 2f. The geopolymer matrix bonded the crystallites together. The existence of crystal was caused by the longer curing age of the fly ash geopolymer at room temperature. The anorthite crystal was indicated by the transformation of fly ash cenosphere as shown in Figure 3. The spherical transformation and denser surface were attributed to the necking reaction between particles (Figure 2f). The presence of crystal was thought to improve the compressive strength of the geopolymer as obtained in Section 3.1.

### 3.3. XRD and u-XRF

Figure 4 illustrates the XRD pattern of the fly ash geopolymer mortar at 28 days and 10 years curing age. M: mullite (Al_6_Si_2_O_12_), Q: quartz (SiO_2_), and A: anorthite (CaAl_2_SiO_8_) are the crystalline phase identified. The XRD diffractogram for the samples of 28 days curing age (Figure 4a) demonstrated an apparent wider amorphous halo. This might be due to the reorganization of the polymer chain and microstructure in the binder and the formation of semi-crystalline structure over 28 days. As shown in Figure 4a, the samples contained mullite and quartz as a major and minor phase, respectively. Mullite and quartz intensities rapidly decreased with longer days of curing age and eventually disappeared after 10 years.

In contrast, the intensity anorthite crystal gradually increased and after 10 years, showing that it can be totally produced after longer curing age. As a result, according to Equation (1) [22], the transformation of mullite into anorthite was completely by the following solid-state reactions:Al_6_Si_2_O_13_ (mullite) + 6CaO + SiO_2_ (Quartz) → 3Ca_2_Al_2_SiO_7_ (gehlenite)3Ca_2_Al_2_SiO_7_ (gehlenite) + Al_6_Si_2_O_13_ + 7SiO_2_ (mullite) → 6CaAl_2_Si_2_O_8_ (anorthite)(1)

The fly ash geopolymer mortar was further examined using synchrotron micro-XRF mapping to gain a better understanding of the element distribution and the possibility for mullite and anorthite development of the curing age samples. Figure 5 depicts the localised area and micro-XRF mapping in the Al–Si–Ca–Fe of the fly ash geopolymer mortar, indicating that Al, Si, and Ca are predominantly located within the samples. The distribution of major elements, particularly light elements that may reflect mineral existence within the geopolymer structure, may be determined using synchrotron micro-XRF. For each distribution element in the integrated region, the colours red, green, and blue signify high, medium, and low intensities, respectively.

Figure 5a–d display the distribution of Si and Al map in geopolymer mortar of 28 days curing age which allowed for the identification of the geopolymer main chain (Si-O-Al/Si) and mullite minerals [23]. The Si represented the medium concentration (green) of Si element for the geopolymers, as demonstrated well by the sample homogeneity. Quartz grains are shown by the high concentration Si area (red). Furthermore, the localized high concentration of the Ca region (Figure 5c) is indicated by lime grains. The longer curing time resulted in considerable changes in the Al, Si, and Ca elemental distribution, resulting in the production of a stable crystalline phase and the significant formation of anorthite as shown in Figure 4b. After curing age up to 10 years, it can be visualised that Al, Si, and Ca are of high intensity in a localised area, reflecting the existence of an anorthite phase. The Ca-rich crystalline minerals are thought to contribute to the glassy and microstructure phase appearance [9]. In other previous studies, Zhang et al. [24], stated that the promotion of crystalline anorthite is due to the higher temperature exposure, while this current work obtained that the crystalline evolution of anorthite from mullite was attributed to the longer curing age.

### 3.4. XTM

Figure 6 depicts the total porosity and pore distribution of fly ash geopolymer mortar at 28 days and 10 years curing age. The majority of the pores in both samples were open, with only a few closed pores. The air occupancy in the sample was used to compute the total porosity, pore diameter, and pore volume. As depicted in Figure 6a, the total porosity of the 28 days curing age sample was 26.45% (25.89% of open pores and 0.56% of closed pores). The pore diameter of 28 days curing age was distributed in the range of 0–5 µm, and its pore volume was 51,353 µm^3^. According to Figure 4b, the total porosity of 10 years curing age sample was 11.04% (8.72% of open pores and 2.32% of closed pores). The pore diameter of the 10 years curing age sample was spread in the range of 0–5 µm, and its pore volume was 11,773.7 µm^3^. In comparison, the 10 years curing age samples had lower overall porosity, pore diameter, and pore volume than the 28 days curing age sample. The reduction in total porosity is due to longer evaporation and dehydroxylation in the geopolymer during the curing period, which generates an isolated pores network and yields a denser surface [11]. This observation is consistent with the finding of surface morphology (Figure 2).

## 4. Conclusions

The long-term performance of fly ash geopolymer material was evaluated after 10 years of curing age at room temperature. The following conclusions can be drawn from the test results:The compressive strength for FA10Y was high compared to FA28D. Compared to FA28D, the compressive strength of FA10Y was approximately ~74% higher than the compressive strength of FA28D.There are increasing densification surfaces and less total porosity within FA10Y compared to FA28D. The columnar anorthite crystal was obtained inside the fly ash spherical particles which contributed to the strength development.The synchrotron micro X-ray tomography revealed a lower range (5–20 µm) of pore distribution and higher localized Al, Si, and Ca elements after 10 years of curing age.

## Figures and Tables

**Figure 1 materials-16-01096-f001:**
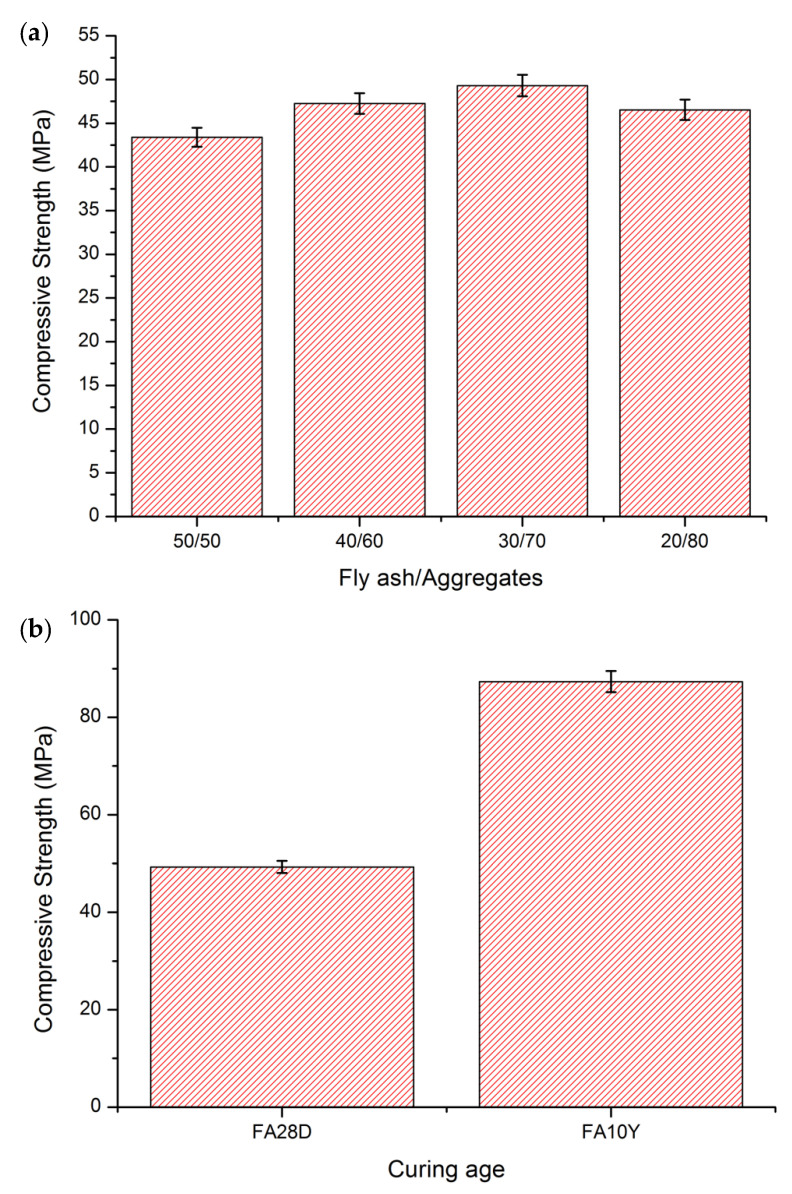
Compressive strength of fly ash geopolymer samples: (**a**) after 28 days curing at various fly ash/aggregates; and (**b**) at 28 days and 10 years of curing age.

**Figure 2 materials-16-01096-f002:**
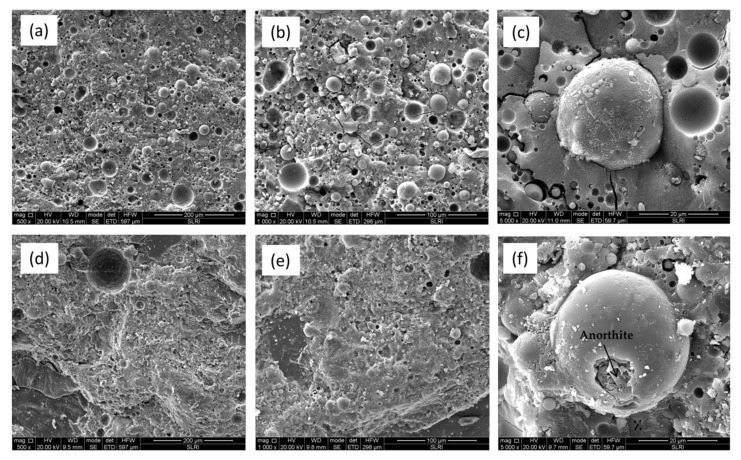
SEM micrograph of fly ash geopolymer at different curing age; 28 days with various magnification: (**a**) 500×; (**b**) 1000×; (**c**) 5000×; and 10 years: (**d**) 500×; (**e**) 1000×; (**f**) 5000×.

**Figure 3 materials-16-01096-f003:**
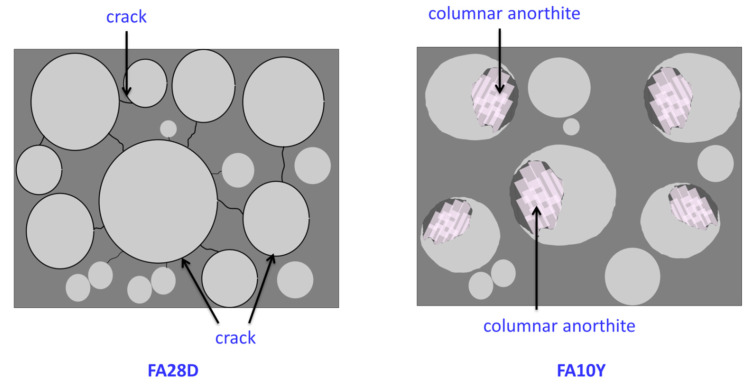
The transformation of fly ash particle after 28 days and 10 years of curing age.

**Figure 4 materials-16-01096-f004:**
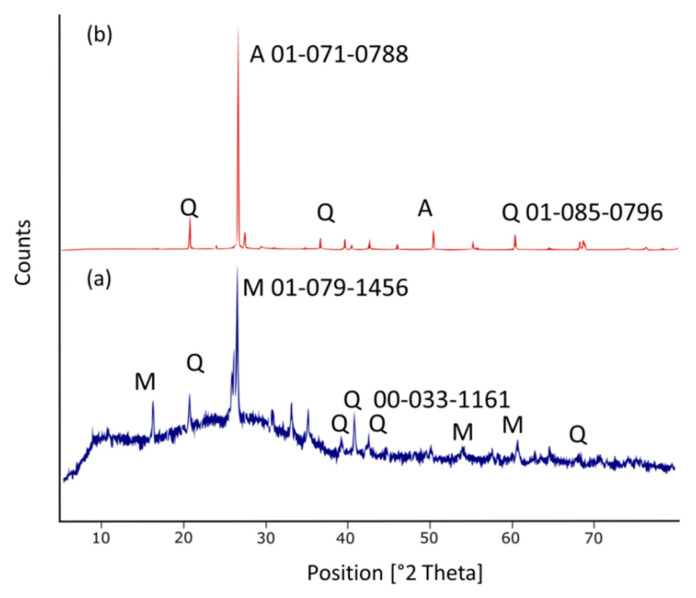
XRD diffractogram of fly ash geopolymer mortar at different curing age: (**a**) 28 days; and (**b**) 10 years.

**Figure 5 materials-16-01096-f005:**
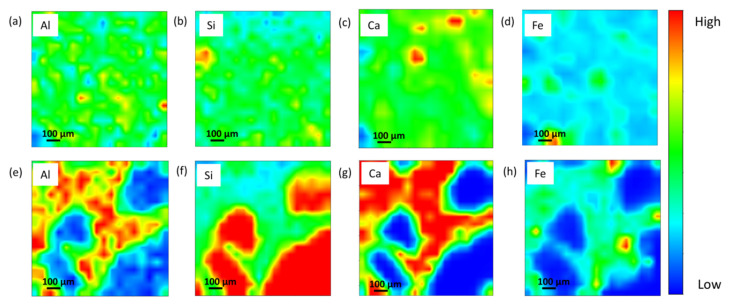
Micro-XRF elemental distribution maps of Al, Si, Ca and Fe of fly ash geopolymer mortar at 28 days (**a**–**d**) and 10 years (**e**–**h**) curing age.

**Figure 6 materials-16-01096-f006:**
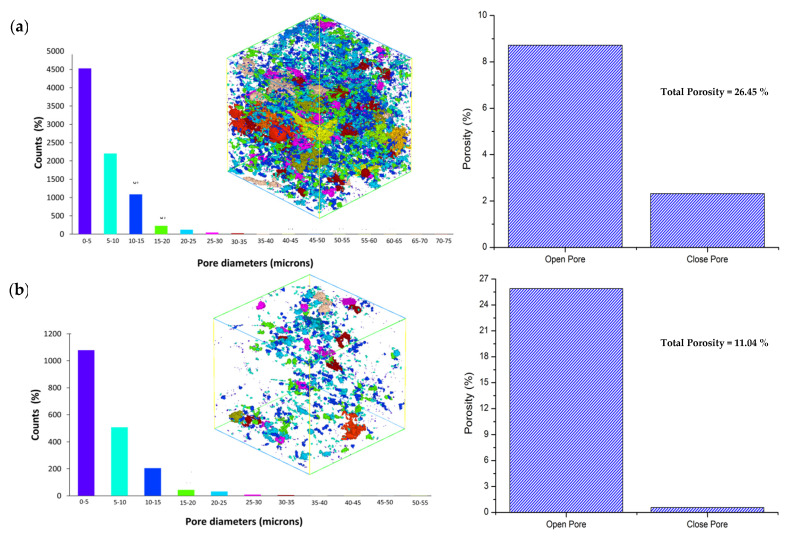
Various porosity size distribution at different curing age: (**a**) 28 days; and (**b**) 10 years of fly ash geopolymer.

**Table 1 materials-16-01096-t001:** Chemical composition of Class C fly ash.

Elemental Oxide (wt%)									
SiO_2_	Al_2_O_3_	CaO	Fe_2_O_3_	TiO_2_	MgO	K_2_O	Na_2_O	P_2_O_5_	LOI
26.4	9.25	21.6	30.13	3.07	0.78	2.58	0.42	1.31	3.02

## Data Availability

Not applicable.

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
