# Peer review of "Mechanical Performance, Microstructure, and Porosity Evolution of Fly Ash Geopolymer after Ten Years of Curing Age"

_materials, 2023, doi:10.3390/ma16031096_

Round 1

Reviewer 1 Report

When I read the title of 10-year geopolymer, I think that this paper can be accepted. However, authors should pay attention to some details. Line 59,54,66-68. Curing conditions, pls give detailed information about curing. If authors conducted other tests, I suggest that they can be added. This is a creditable work. The detailed problems are as followed:

1.Line 2, porosityevolution?

2.Line34,materialhas

3.Line37,improve

4.Line41, give details for supporting this idea, I think OPC is cheaper that geopolymer, activator is expensive and CO2 emission is great.

5.Line49,ensuring

6.Line54,structure

7.Line61,harden

8.Line106,m

9.Line155,Strengthmeasurement?

10.Line157,thatthe

11.Line181,particle

12.Line193, The

13.Line243-244,yielding

14.Line268, localised viscous sintering, pls explain it carefully, currently, I believe in that it is difficult to understand

15.We can’t know how to curing, thus more description is needed.

Regards,

Author Response

  1. When I read the title of 10-years geopolymer, I think that this paper can be accepted. However, authors should pay attention to some details. Line 59, 54, 66-68. Curing conditions, please give detailed information about curing. If authors conducted other test, I suggest that they can be added. This is a creditable work. The detailed problems are as followed

Response: Thank you for your constructive suggestion. To address the reviewer’s concern, we have been clearly revised the manuscript as followed the comment given.

  1. Line 2, porosityevolution?

Response: We are grateful to the reviewer for noticing this. We have made extensive editing for the writing error. (Line 2-3, Page 1)

  1. Line 34, materialhas

Response: We are grateful to the reviewer for noticing this. We have made extensive editing for the writing error. (Line 33, Page 1)

  1. Line 37, improve

Response: We gratefully appreciate for your valuable suggestion. To avoid any misunderstanding, the sentence has been removed in the revised manuscript.

  1. Line 41, give details for supporting this idea, I think OPC is cheaper that geopolymer, activator is expensive and CO2 emission is great.

Response: We are sorry for the inconvenience brought to the reviewer. We apologize for our inaccurate description. To avoid any misunderstanding, the sentence has been removed in the revised manuscript.

  1. Line 49, ensuring

Response: Thank you for the Reviewer’s suggestions. We have made the corresponding revision, replacing the correct word. (Line 45, Page 1)

  1. Line 54, structure

Response: Thank you for your kind suggestion, which is highly appreciated. We have made the corresponding revision. (Line 49-51, Page 2)

  1. Line 61, harden

Response: Thank you for your kind suggestion, which is highly appreciated. We have made the corresponding revision, replacing the correct word. (Line 58, Page 2)

  1. Line 106, m

Response: Thank you for your kind suggestion, which is highly appreciated. We have made the corresponding revision. (Line 102, Page 3)

  1. Line 155, Strengthmeasurement?

Response: Thank you for your kind suggestion, which is highly appreciated. We have made the corresponding revision. (Line 164, Page 4)

.  

  1. Line 157, thatthe

Response: Thank you for your kind suggestion, which is highly appreciated. We have made the corresponding revision.  (Line 166, Page 4)

  1. Line 181, particle

Response: Thank you for your kind suggestion, which is highly appreciated. We have made the corresponding revision, replacing the correct word. (Line 190, Page 5)

  1. Line 193, The

Response: Thank you for your kind suggestion, which is highly appreciated. We have made the corresponding revision. (Line 202, Page 6)

  1. Line 243-244, yielding

Response: Thank you for your kind suggestion, which is highly appreciated. We have made the corresponding revision, replacing the correct word. (Line 253-256, Page 8)

  1. Line 268, localised viscous sintering, please explain it carefully, currently, I believe in that it is difficult to understand

Response: We are sorry for the inconvenience brought to the reviewer. We apologize for our inaccurate description. To facilitate your understanding, we have rewritten the sentence for better flow and continuity. (Line 278-279, Page 9)

  1. We can’t know how to curing, thus more description is needed.

Response: We are sorry for the inconvenience brought to the reviewer. We apologize for our inaccurate description. To facilitate your understanding, we have rewritten the sentence for better flow and continuity. (Line 105-110, Page 3).

Reviewer 2 Report

The work is well written and well done, but it feels like a didactic exercise. In fact it leads to obvious conclusions and therefore the work does not appear very interesting as it does not lead to new knowledge. It has the only quality of demonstrating what is already expected even if with innovative techniques and on geopolymeric materials.

  Two critical notes.

Statements such as "geopolymer has the potential improve the concrete industry by lowering the carbon footprint" or "geopolymer manufacture is less expensive than OPC concrete" are indemonstrable and even the reference [13] does not satisfy in this sense.

Comparing samples prepared 10 years ago with fresh samples is a good way not to throw away previous activities, but in the space of ten years the raw materials can be changed, certainly the operators, the synthesis protocols and many other things: the level of detail especially with respect to the evaluation of porosity it seems really excessive if not out of place.

Author Response

  1. The work is well written and well done, but it feels like a didactic exercise. In fact, it leads to obvious conclusion and therefore the work does not appear very interesting as it does not lead to new knowledge. It has the only quality of demonstrating what is already expected even if with innovative technique and on geopolymeric materials.

Response: We highly appreciated the reviewer’s positive appraisal of our manuscript.

  1. Statements such as “geopolymer has the potential improve the concrete industry by lowering the carbon foorprint” or “geopolymer manufacture is less expensive than OPC concrete” are indemonstrable and even the reference [13] does not satisfy in this sense.

Response: We are sorry for the inconvenience brought to the reviewer. We apologize for our inaccurate description. To avoid any misunderstanding, the sentence has been removed in the revised manuscript.

  1. Comparing samples prepared 10 years ago with fresh samples is a good way not to throw away previous activities, but in the space of ten years the raw materials can be changes, certainly the operators, the synthesis protocols and many other things: the level of details especially with respect to the evaluation of porosity it seems really excessive if not out of place.

Response: Thank you for your considerable comments. The details of the synthesis protocol have been clearly explained in Section 2. We have rewritten the methodology for better flow and continuity. (Line 105-110, Page 3) and (Line 131-147, Page 4)

Reviewer 3 Report

The research data used in this article is invaluable and deserves recognition. However, in the article, some issues still need to be corrected or discussed.

1.          Page 3, line 106, “poured into 100 mm x 100 mm x 100 m cubic plastic moulds”, is it correct? Or should it be 100 mm x 100 mm x 100 mm?

2.          The title of Chapter 3.1 “Compressive Strengthmeasurement”, it should be “Compressive Strength measurement”, please correct it.

3.          Page 4, line 157, what is “thatthe”?

4.          The correlation between fly ash geopolymer aggregates and fly ash/aggregates ratios should be clearly pointed out in the article.

5.          What are the curing conditions for 10 years of curing?

6.          About SEM results, how to prove that the cracks in the specimen are not caused by taking samples for SEM? Is it possible to observe the specimen with optical microscope?

7.          According to the XRD results, Fly-ahs based geopolymer after curing for 10 years, the amorphous amount is increased. However, if the crystalline phase transfers to the amorphous phase, it means that Fly-ahs based geopolymer is unstable, which might cause lots of problems with durability. The author should confirm the experimental results again.

Author Response

  1. The research data used in this article is invaluable and deserves recognition. However, in this article, some issues still need to be corrected or discussed.

Response: We highly appreciated the reviewer’s positive appraisal of our manuscript.

  1. Page 3, Line 106, “ poured into 100mm x 100mm x 100mm cubic plastic moulds”, it is correct? Or should it be 100mm x 100mm x 100mm?

Response: Thank you for your kind suggestion, which is highly appreciated. We have made the corresponding revision. (Line 102, Page 3)

  1. The title of Chapter 3.1 “ Compressive Strengthmeasurement”, it should be “Compressive Strength measurement”, please correct it.

Response: Thank you for your kind suggestion, which is highly appreciated. We have made the corresponding revision. (Line 164, Page 4)

  1. Page 4, Line 157, what is “that the”?

Response: Thank you for your kind suggestion, which is highly appreciated. We have made the corresponding revision. (Line 166, Page 4)

  1. The correlation between fly ash geopolymer aggregates and fly ash/aggregates ratios should be clearly pointed out in the article.

Response: We thank the reviewer for the valuable suggestion. The various fly ash/aggregates ratios are to determine which ratios will produce the optimum strength. Hence, the selected optimum strength will be used for 10 years of curing samples (FA10Y) to compare the current samples (FA28D). This was clarified in Section 3.1. (Line 175-177, Page 4)

  1. What are the curing conditions for 10 years of curing?

Response: Thank you for your considerable comments. We have rewritten the methodology for better flow and continuity. (Line 105-110, Page 3).

  1. About SEM results, how to prove that the cracks in the specimen are not caused by taking samples for SEM? Is it possible to observe the specimen with optical microscope?

Response: Thank you for the great suggestions. The justification of crack existence in the specimen is not caused by taking a sample for SEM was highlighted in the revised manuscript. Also, the crack visualization was impossible to be observed via an optical microscope. (Line 120, Page 3)

  1. According to the XRD results, Fly ash based geopolymer after curing for 10 years, the amorphous amount is increased. However, if the crystalline phase transfer to the amorphous phase, it means that fly ash based geopolymer is unstable, which might cause lots of problems with durability. The author should confirm the experimental results again.

Response: We are sorry for the inconvenience brought to the reviewer. We apologize for our inaccurate description. The sentence has been revised and replaced with “The XRD diffractogram for the samples of 28 days curing age (Figure 4a) demonstrated an apparent wider amorphous halo. (Line 225-226, Page 7). Otherwise, the evolution of crystallization from mullite to anorthite has been supported by the Eq.1 [23]. Below is the detail of the previous study.

[23]  Qin, J., Cui, C., Cui, X., Hussain, A., Yang, C. and Yang, S., 2015. Recycling of lime mud and fly ash for fabrication of anorthite ceramic at low sintering temperature. Ceramics International, 41(4), pp.5648-5655.

Reviewer 4 Report

This manuscript compares of mechanical and structural properties of selected geopolymer after 28 days and 10 years. The work has its value mainly due to the long storage time of the samples.

I would recommend following points to the attention of the authors:

– Many spaces are missing in the text - it should be edited correctly.

Unfortunately, I cannot see the anorthite in Figure 2f - it should be enlarged so as to show the described structures.

– In my opinion, Figure 3 is redundant.

– The peaks in the diffraction patterns are ambiguous. Please review these results again.

– Figure 5 is missing a scale. It is not known what area of the sample was depicted.

The compressive strength of FA10Y was approximately ~74% higher than the 279 compressive strength of FA28D – This value is not valid.

The above observations do not allow the reviewer to recommend the paper in its current form for publication.

Author Response

  1. This manuscript compare of mechanical and structural properties of selected geopolymer after 28 days and 10 years. The work has its value mainly due to the long storage time of the samples.   

Response: We highly appreciated the reviewer’s positive appraisals of our manuscript.

  1. Many spaces are missing in the text- it should be edited correctly.

Response: Thank you for your constructive suggestion. To address the reviewer’s concern, we have clearly revised the manuscript. All the changes were been highlighted in yellow color in the revised manuscript.

  1. Unfortunately, I cannot see the anorthite in Figure 2f, it should be enlarged so as to show the described structures,

Response: We thank the reviewer for this suggestion. To facilitate a better understanding, the anorthite has been labeled as shown in Figure 2f. Also, the existence of anorthite has been supported by previous studies, “…columnar anorthite crystal and plerospheres of thin-walled spherical fly ash [23] can be clearly seen in Figure 2f.”

  1. In my opinion, Figure 3 is redundant.

Response: We are sorry for the inconvenience brought to the reviewer. We apologize for our inaccurate description. Figure 3 is a schematic diagram that shows the existence of anorthite in the sphere of fly ash. It has been referred to in Figure 2f which believes that the anorthite was contribute to the denser surface and phase crystallization.

  1. The peaks in the diffraction patterns are ambiguous, Please review the result again.

Response: We are sorry for the inconvenience brought to the reviewer. We apologize for our inaccurate description. The peaks in the diffraction were present in Section 3.3 XRD analysis. The result shows that the halo amorphous peak was obtained after 28 days of curing samples. Then, the evolution of crystallization was obtained for 10 years of curing samples. The main peak was identified with the formation of anorthite from mullite. Also, this was been proved with the previous studies via Eq. 1 [23].

  1. Figure 5 is missing a scale. It is not known what area of the samples was depicted.

Response: We are sorry for the inconvenience brought to the reviewer. We apologize for our inaccurate description. The scales were added in Figure 5.

  1. The compressive strength of FA10Y was approximately ~74% higher than the 279 compressive of FA28D- This value is not valid.

Response: We are sorry for the inconvenience brought to the reviewer. We apologize for our inaccurate description. The result of compressive strength was compared between the FA28D and FA10Y of samples. The higher strength was obtained for FA10Y due to the crystallization during curing age. Also, the result of strength was supported by the SEM and XTM results which formed a denser and less pore structure.

Round 2

Reviewer 4 Report

I believe that my comments were mostly received and implemented. Therefore, please consider the manuscript for publication.

Author Response

Thank you for your valuable comment. We highly appreciated the reviewer’s positive appraisal of our manuscript.